# How social relationships shape moral wrongness judgments

Brian D. Earp [1]✉, Killian L. McLoughlin[1], Joshua T. Monrad [1], Margaret S. Clark [1] & Molly J. Crockett [1]✉

Judgments of whether an action is morally wrong depend on who is involved and the nature of their relationship. But how, when, and why social relationships shape moral judgments is not well understood. We provide evidence to address these questions, measuring cooperative expectations and moral wrongness judgments in the context of common social relationships such as romantic partners, housemates, and siblings. In a pre-registered study of 423 U.S. participants nationally representative for age, race, and gender, we show that people normatively expect different relationships to serve cooperative functions of care, hierarchy, reciprocity, and mating to varying degrees. In a second pre-registered study of 1,320 U.S. participants, these relationship-specific cooperative expectations (i.e., relational norms) enable highly precise out-of-sample predictions about the perceived moral wrongness of actions in the context of particular relationships. In this work, we show that this 'relational norms' model better predicts patterns of moral wrongness judgments across relationships than alternative models based on genetic relatedness, social closeness, or interdependence, demonstrating how the perceived morality of actions depends not only on the actions themselves, but also on the relational context in which those actions occur.

[1] Department of Psychology, Yale University, New Haven, CT, USA. Shared senior authorship: Margaret S. Clark, and Molly J. Crockett.
✉email: brian.earp@yale.edu; molly.crockett@yale.edu

Moral psychology has been dominated by studies of judgments and behaviors concerning strangers: individuals who stand in no particular relationship to one another, and who may or may not interact in the future[1]. Researchers conducting such studies commonly ask participants to make decisions that impact anonymous others[2–4] or to judge the moral acceptability of hypothetical actions taken by thinly-described agents, as in sacrificial dilemmas where participants must judge the permissibility of killing one person to save a greater number[5–8]. Of course, people often do encounter strangers as they go about their lives, and the interpersonal standing implied by such encounters can be seen as a bare-bones social relationship involving certain minimal obligations: for example, a "duty of easy rescue" in the case of emergencies[9]. The copious research on moral judgments in the context of stranger-stranger relationships thus sheds important light on at least one important aspect of our moral psychology.

However, the vast majority of our moral judgments in everyday life do not concern strangers. Rather, they concern familiar others with whom we stand in particular, often ongoing relationships[10]. The stakes of such moral judgments for the maintenance of our personal social networks typically are higher than the stakes of analogous judgments pertaining to strangers. Moreover, moral judgments about interactions between strangers often will differ in systematic ways from judgments about interactions between friends, family members, or other familiar individuals in the same situation[11–13]. For example, consider someone who could easily feed a hungry individual but fails to do so. If this person is a mother failing to feed her own child, she likely will be seen as highly blameworthy. But if the person is a local restaurant owner failing to feed a non-paying customer, the same behavior likely will not be seen as blameworthy under ordinary conditions[14].

A number of theorists have highlighted relational context as likely to be important for understanding moral judgment and behavior[10,12,15–18]. In line with these developments, there is now a small but growing empirical literature which explores how moral judgments of particular actions vary across different types of social relationships[19–34]. How these relationships are theorized depends on the study. For example, one recent study characterized relationships in terms of the genetic relatedness of the interaction partners, and showed how varying this factor affects moral judgments about helping behavior[35]. Another recent study characterized relationships in terms of the authors' intuitive sense of the social closeness and relative interdependence of the interaction partners – regardless of genetic relatedness – and tested the influence of these factors on judgments about violations of care[36]. Researchers also have sought to predict moral wrongness judgments of actions in relational context from a single cooperative function thought to characterize a given relationship (e.g., care for a sibling relationship, hierarchy for a teacher-student relationship, and so on)[28].

These studies demonstrate that moral judgments of one and the same action often differ across different types of relationships,

depending on how relationship "type" is understood. What is missing, however, is a systematic, data-driven account of the multiple cooperative functions that can characterize any given social relationship[14], and an explicit comparison of how well such cooperative functions predict relationally-situated moral judgments relative to alternative models such as genetic relatedness, social closeness, and interdependence. We aim to fill that gap with the present research.

In contrast to genetic relatedness, which can be determined objectively, and the constructs of social closeness and interdependence, both of which have been carefully defined within the relationship science literature, there is no agreed-upon set of cooperative functions prescribed for different social relationships to solve characteristic coordination problems. Recognizing both the theoretical overlap and diversity among the various existing taxonomies of cooperative functions[16,37,38], we build on work by Bugental[39]. This work describes a distinctive set of cooperative functions that serve to coordinate behavior in interpersonal relationships. Each function represents an efficient, socially acceptable solution to a particular type of recurrent coordination problem[37], enabling cooperation partners to mutually benefit over repeated interactions[38,40,41]. We focus here on four cooperative functions that solve dyadic or two-party coordination problems: care, reciprocity, hierarchy, and mating (Table 1).

As has been noted previously, any given relationship may serve multiple cooperative functions, either characteristically or in a specific context[16,39]. We propose that within a given society, there are prescriptive norms for the set of cooperative functions different relationships should serve ('relational norms'). In the present work, we sought to (i) describe patterns of relational norms for a large set of common dyadic relationships in a U.S. cultural context; (ii) use these patterns of relational norms to predict out-of-sample judgments of moral wrongness for actions that violate those norms across relationships; and (iii) to compare this 'relational norms' model with alternative ways of characterizing dyadic relationships, i.e., in terms of genetic relatedness, social closeness, and interdependence.

With respect to aims (i) and (ii), we predicted that relational norms would robustly predict moral judgments about the wrongness of actions in relational context. The basis for this prediction is straightforward: the more a particular set of cooperative functions matters for a given relationship, the morally worse it should be judged to be to neglect or frustrate those same functions within that relationship. Because relationships vary in terms of the set of cooperative functions they are normatively expected to serve (and to what degree), a given action may be judged to be seriously wrong in the context of one relationship but entirely acceptable in the context of another.

We further predicted that our relational norms model would better explain the variance in moral wrongness judgments across relationship dyads than genetic relatedness, social closeness, and interdependence, which we believe offer incomplete predictive accounts of such judgments in relational context. To illustrate,

---

**Table 1 Cooperative functions of dyadic relationships, adapted from Bugental[39].**

| Cooperative function | Coordination problem to be solved |
|---|---|
| Care | Securing basic welfare needs through non-contingent provision (or acceptance) of help or support; maintaining safety; encouraging learning |
| Reciprocity | Coordinating behavior between individuals with functionally similar (or equal) status, power, authority, or claim on a resource |
| Hierarchy | Coordinating behavior between individuals with different (unequal) status, power, authority, or claim on a resource |
| Mating | Finding and maintaining sexual partners; ultimately, producing and ensuring the survival of offspring |

We note that the care function is based on the work of Clark and colleagues concerning "communal" relationships[59]; it conceptually overlaps with, and replaces, the "attachment" function in Bugental's model. Because our model is focused on dyadic interactions, we also do not include Bugental's group-level "coalition" function in this table (see Supplement Section 1.4.3. for data pertaining to the coalition function).

imagine that Person A fails to behave in a deferential manner toward Person B. Insofar as the relationship is normatively expected to be governed by the hierarchy function (see Table 1), with Person A in the subordinate position, such behavior likely will be judged as morally wrong. By contrast, consider how genetic relatedness might explain the wrongness of this action. Some genetically close relationships, such as the parent-child relationship, may, indeed, normatively rely on the hierarchy function to coordinate behavior, and to the extent they do, the action might be judged to be wrong. However, other genetically close relationships, such as siblings of a similar age, are less likely to rely on the hierarchy function, while some genetically distant relationships, such as a typical boss-employee relationship, might be equally or even more likely to rely on the function. Thus, genetic relatedness ultimately may prove to be largely independent of the question of what makes certain actions liable to be judged morally wrong.

In summary, unlike most prior work in moral psychology, which has been designed to predict moral judgments from features of actions regardless of who performs the action or their relationship to the affected other, here we consider features of common social relationships that we predict will shape moral judgments of actions that occur in the context of specific relational dyads. We show that the similarity between relationship dyads in terms of their prescribed cooperative functions—or relational norms—corresponds to similarity in moral wrongness judgments between relationships. Put another way, dyads with similar relational norms within a given society are associated with similar patterns of moral wrongness judgments across actions, whereas dyads with dissimilar relational norms are associated with divergent patterns of moral wrongness judgments across actions. Finally, we show that relational norms more strongly predict patterns of moral wrongness judgments across relationships than alternative predictors, including genetic similarity, social closeness, or interdependence.

## Results

**Relational norms vary across common dyadic relationships.** We first measured relationship-specific patterns for prescribed cooperative functions (i.e., relational norms) for a set of common dyadic relationships in the U.S. (study design, sampling plan, and exclusion criteria pre-registered at aspredicted.org, #26400). Participants (final $n = 423$, U.S. nationally representative for age, race, and gender; "Sample 1") rated 20 common dyads on the extent to which each is normatively expected to serve the functions of care, reciprocity, hierarchy, mating, and coalition. Results for all 20 relationships across the four functions from Table 1 are depicted in Fig. 1 (for coalition data, see Supplement Section 1.4.3.).

As can be seen in Fig. 1, relational norms varied markedly across dyads in several respects. The reciprocity function generally was prescribed for most dyads ($M$ across dyads = 54.23, $SD = 49.64$; higher than the scale midpoint with a Bonferroni corrected alpha = .0125, $t(8,459) = 100.47$, $p < .001$, $d = 1.09$; note that all tests reported in the manuscript are two-sided). Meanwhile, the mating function was negatively prescribed (i.e., proscribed) for most dyads ($M$ across dyads = −63.02, $SD = 62.01$; lower than the scale midpoint with the same correction, $t(8,459) = 93.48$, $p < .001$, $d = 1.02$), with a few obvious exceptions (romantic partners, $M = 95.12$, $SD = 12.94$; friends-with-benefits, $M = 58.43$, $SD = 51.21$). Participants demonstrated higher levels of agreement about whether dyads were expected to serve the mating ($SD_{mean}$ across dyads = 32.26) and care ($SD_{mean}$ across dyads = 37.82) functions, relative to the reciprocity ($SD_{mean}$ across dyads = 42.25) and hierarchy ($SD_{mean}$ across dyads = 53.72) functions.

Figure 2 depicts the four-dimensional relational norm profiles (i.e., sets of prescribed cooperative functions) for a subset of relationships studied, and illustrates several additional features of our data (see Supplement Section 1.4.4. for functional profile plots for all 20 relationships). First, some relationships are highly functionally "polarized," showing substantial deviation in mean prescriptions across the four cooperative functions, with at least one function anchored at an extreme end of the scale. An example is the parent and under-18 child relationship ($SD$ across cooperative functions = 85.33, 85.2 for the mother-child and father-child relationship, respectively), which is characterized by a strongly positive expectation for care and a strongly negative expectation for mating. By contrast, other relationships are less functionally polarized, such as the relationship between strangers ($SD$ across functions = 37.93). In these relationships, prescribed cooperative functions are relatively evenly spread across the measured spectrum.

Second, some relationships are functionally "specific," that is, they are only strongly expected to serve a single cooperative function. For example, the roommate/housemate relationship is strongly expected to serve the reciprocity function ($M = 87.30$, $SD = 21.71$), but less so the care ($M = 24.9$, $SD = 43.64$), hierarchy ($M = −4.48$, $SD = 63.00$), and mating functions ($M = −52.39$, $SD = 49.85$). Similarly, the boss-employee relationship is strongly expected to serve the hierarchy function ($M = 84.75$, $SD = 24.68$), but less so the reciprocity ($M = 29.14$, $SD = 58.93$) or care functions ($M = 7.86$, $SD = 50.21$), and the mating function not at all ($M = −92.17$, $SD = 23.98$). By contrast, other relationships are functionally "pluralistic," that is, they are strongly expected to serve multiple cooperative functions. A key example is the romantic partner relationship ($M$ across functions = 64.61), which is strongly expected to serve three of the four cooperative functions: care ($M = 92.43$, $SD = 17.06$), mating ($M = 95.12$, $SD = 12.92$), and reciprocity ($M = 84.95$, $SD = 27.28$) but not, in this sample from the United States, the hierarchy function. See Supplement Sections 1.4.1. and 1.4.2. for the rankings of all 20 relationship dyads on the dimensions of polarization and specificity.

We also find gender differences in prescribed cooperative functions across relationships. After scaling the raw scores to each participant's mean rating, we built a mixed linear effects regression model controlling for relevant demographic information (income, religiosity, and political orientation entered as fixed effects), including participant and relationship dyad type as random effects. With a Bonferroni correction (alpha = .0125 for each of the following effects), the model revealed that women ($M = 0.43$, $SD = 0.75$), compared to men ($M = 0.37$, $SD = 0.79$), reported stronger average expectations that relationships will serve a function of care ($p < .001$, 95% CI [.03, .10]), consistent with the existing literature[42–45]. This divergence was most apparent for the roommate/housemate ($M_{female} − M_{male} = 0.17$), customer-seller ($M_f − M_m = 0.15$), teacher-student ($M_f − M_m = 0.14$), neighbor ($M_f − M_m = 0.14$), and colleague/classmate ($M_f − M_m = 0.13$) relationships. Regarding mating, the opposite pattern was found, also consistent with the existing literature[46,47]. Men ($M = −1.12$, $SD = 0.97$), compared to women ($M = −1.2$, $SD = 0.89$), reported stronger average expectations that relationships will serve a mating function ($p < .001$, 95% CI [.03, .11]). This divergence was most apparent for the friends-with-benefits ($M_m − M_f = 0.27$), roommate/housemate ($M_m − M_f = 0.25$), acquaintance ($M_{male} − M_{female} = 0.24$), close friend ($M_m − M_f = 0.25$), colleague/classmate ($M_m − M_f = 0.2$), stranger ($M_m − M_f = 0.16$), and neighbor ($M_m − M_f = 0.17$) relationships. Additional demographic analyses are reported in Supplement Section 1.4.5.

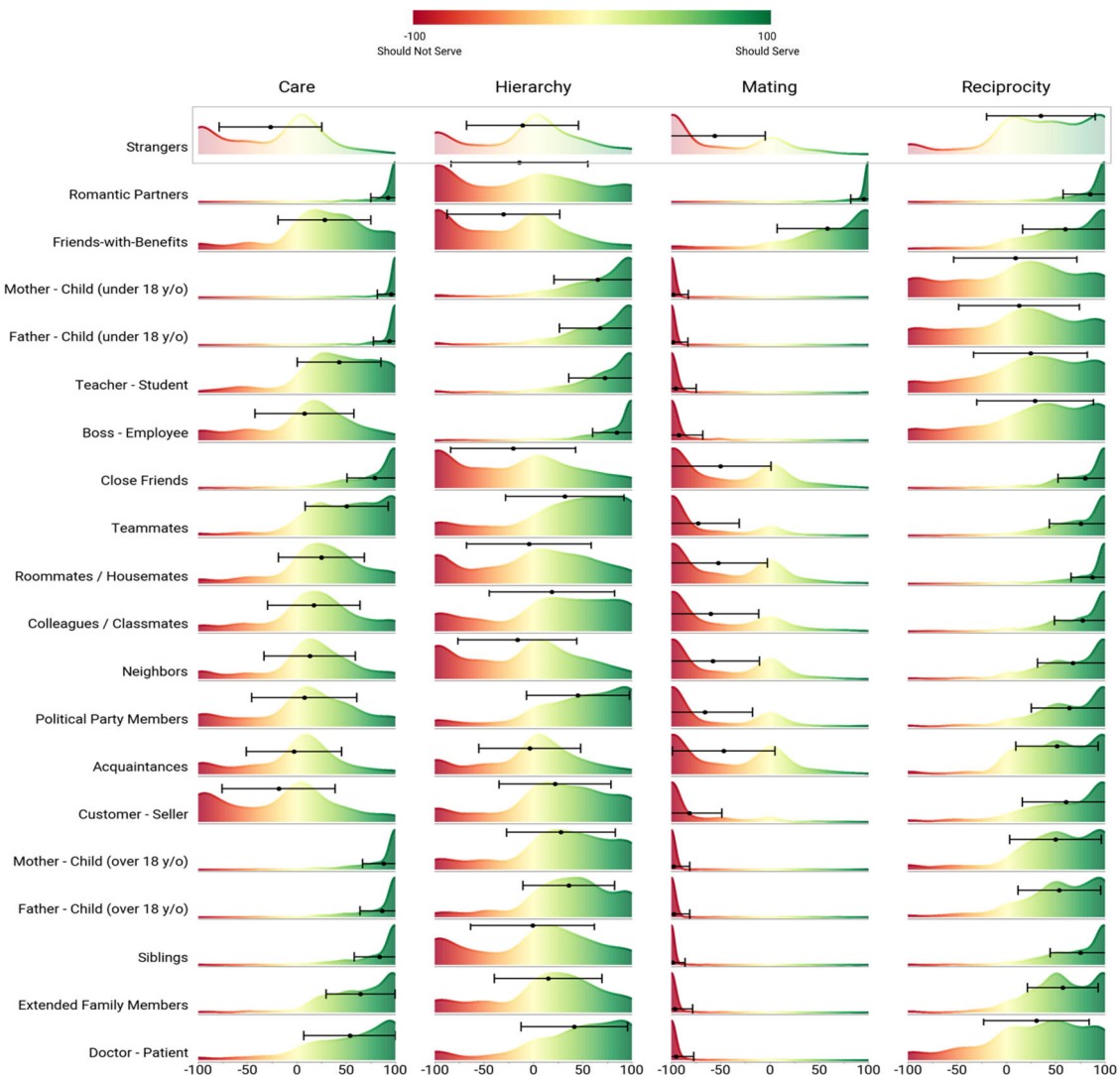

**Fig. 1 Kernel density plots of prescribed cooperative functions for 20 common relationship dyads.** Dots represent the population mean prescription for each cooperative function within each relationship, caps represent +/− one standard deviation. The height of the curve represents density: the likely proportions of scores (relative to each function) that fall within the given range along the x-axis. Source data are provided as a Source Data file.

**Common relationships are hierarchically clustered around relational norms.** Next, we sought to quantify the distinctiveness of each relationship in four-dimensional relational norm space. Because in many instances patterns of prescribed cooperative functions were not normally distributed in our study population (see Fig. 1), characterizing relationship differences in terms of their average relational norm scores would sacrifice considerable information. We therefore calculated the Kolmogorov-Smirnov (K-S) distance statistic (a quantification of the difference in overall shape between any two empirical distributions) for each cooperative function for each possible pair of relationships, and averaged across functions to calculate the overall dissimilarity in relational norms for each relationship pair. This approach is conceptually similar to representational similarity analysis[48], but incorporates information about the shapes of the relational norm distributions in addition to distribution means.

We used the relational norm dissimilarity values to conduct a hierarchical clustering analysis using a farthest-point algorithm: $d(u, v) = \max(\text{dist}(u[i], v[j]))$[49]. This revealed four main clusters, depicted in Fig. 3a, b, which align with intuitive relational categories. The first cluster consists of sexual relationships (romantic partners and friends-with-benefits). The second cluster

consists of hierarchical relationships with highly unequal authority between individuals (parents and their minor children, teacher-student, boss-employee). The third cluster includes relationships characterized largely by reciprocal interactions between equals (e.g., customer-seller, roommates/housemates, strangers). And the fourth, final cluster includes familial or other caring relationships (e.g., siblings, extended family members, parents and their adult children).

Based on these analyses, we identified a subset of 10 relationships with relatively distinctive relational norms (see Supplement Section 2.1. for the selection procedure). This subset included long-term romantic partners, friends with benefits, boss and employee, colleagues or classmates, mother/father and under-18 child, siblings, close friends, roommates or housemates, teammates, and strangers. Relational norm profiles for these relationships are depicted in Fig. 2. We next sought to predict moral judgments of actions performed in the context of these relationships on the basis of their relational norm profiles.

**Relational norm profiles predict relationship-specific moral wrongness judgments out of sample.** To test the hypothesis that

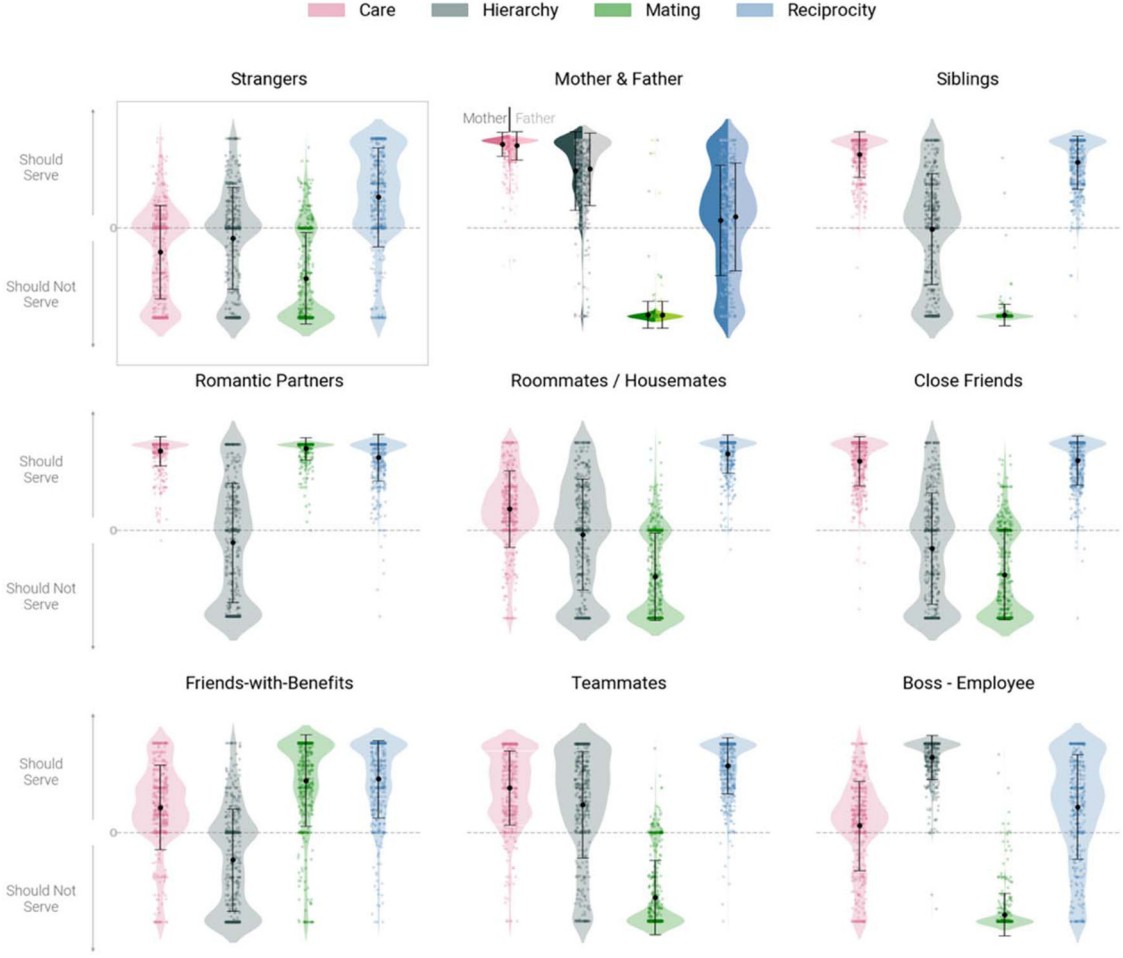

**Fig. 2 Relational norm profiles for a subset of 10 relationships.** Pink represents care, black represents hierarchy, green represents mating, blue represents reciprocity. The raw data ($n = 423$ independent ratings per function per relationship; total $n = 16,920$) are shown in individual dots; error bars represent the mean (dot) and $+/- 1\,SD$ (caps). Note: Mother/Father and under-18 child have been combined into a single plot. Plots for all 20 relationships are in Supplement Section 1.4.4. Source data are provided as a Source Data file.

relational norm profiles would predict patterns of moral judgments across social relationships, we first assembled a set of common behaviors that would plausibly weaken or violate one or more of the cooperative functions. Fifteen trained judges rated 86 action statements of the form "Person A does X to Person B" on the extent to which each described action characteristically would weaken (that is, violate or impair) or strengthen each of the cooperative functions, setting moral questions aside (that is, the judges were instructed not to think about whether an action might be right or wrong in any relationship, but only whether it would weaken or strengthen each function). There was very high interrater agreement in these ratings (ICC(3, k) = .97). Using these data, we selected a final set of 12 characteristic function-weakening action statements, with 3 statements for each of the 4 dyadic functions (see Methods for the algorithm used to select the final sub-set). See Fig. 4.

As can be seen in Fig. 4, each action was rated by the judges as having both a main (i.e., "target") effect on a given function, as well as "side effects" on the other cooperative functions. For example, "Person A sees Person B crying and walks away from them" was rated as most characteristic in weakening the care function ($M = -87.9$, $SD = 15.5$), but also was rated as characteristically weakening the mating function, albeit to a lesser extent ($M = -40.1$, $SD = 35.0$). The fact that one and the same action might simultaneously weaken several cooperative

functions is to be expected, depending on the logic of each function and the nature of the action. To account, then, for the specificity of each action as a function-weakener, we computed a "target specificity" variable (i.e., main effect minus the mean of side effects) for each action for use in subsequent analyses.

Having identified a set of actions, drawn from everyday life, that were judged to characteristically weaken one or more prescribed cooperative functions, our next step was to assess moral judgments concerning those actions in the context of specific relationships. To do this, we recruited a new group of participants (online U.S. convenience sample, final $n = 1,320$; "Sample 2"), after pre-registering our hypothesis, study design, sampling plan, exclusion criteria, and analysis approach at aspredicted.org, #31592. These "naïve" Sample 2 participants were given no information about cooperative functions. Rather, each participant was assigned randomly to consider 1 of the 10 functionally distinctive relationships identified above, and was asked to rate the moral wrongness of all 12 actions listed in Fig. 4 in the context of that relationship (e.g., "Imagine that an employee refuses to follow a reasonable order from their boss. How morally wrong would that be, if at all?"). We also asked participants to rate each action on how likely it would be to occur in real life, in order to be able to control for perceived violations of non-moral (i.e., social-conventional) expectations[50] in a pre-registered secondary analysis (see "action likelihood" variable

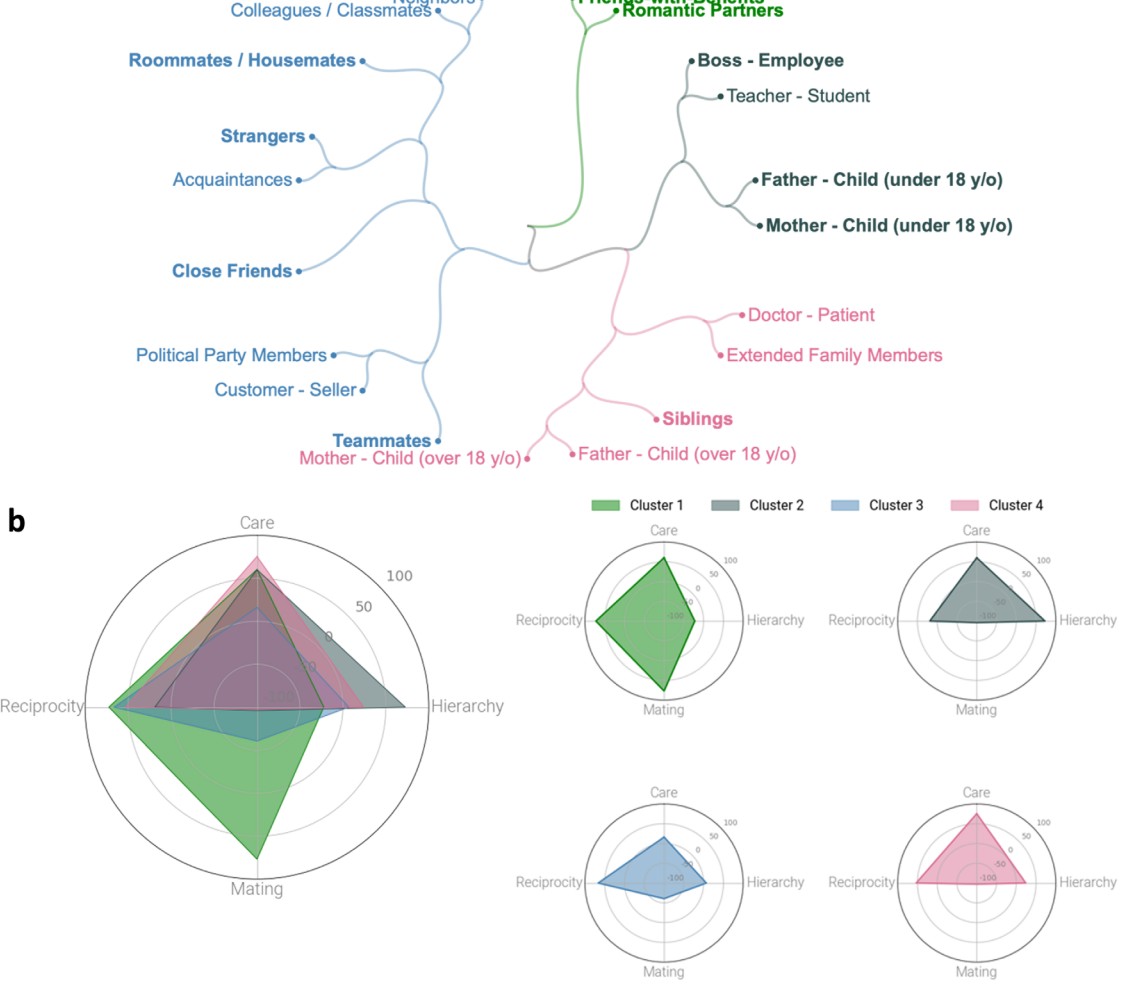

**Fig. 3 Hierarchical clustering of relationships.** Circular dendrogram visually representing the mean Kolmogorov-Smirnov (K-S) distance between relationships in four-dimensional relational norm space, clustered hierarchically according to the Voorhees[49] method (**a**); relationships selected for Study 2 are highlighted in a darker shade. Radar plots derived from the hierarchical cluster model are depicted in the bottom half of the figure (**b**). The left panel shows the overlapping clusters; the right panel shows each cluster on its own set of axes. Source data are provided as a Source Data file.

below). For each participant, we computed the mean moral wrongness rating for each of the four cooperative function-weakening categories within their assigned dyad. See Fig. 5 for distributions of moral wrongness ratings for each function-weakening cateogry for each relationship (for demographic analyses, see Supplement Section 2.5.1.).

We turn now to our main, pre-registered hypothesis. As a first approach, we sought to predict Sample 2 moral wrongness judgments (i.e., for weakening each of the four cooperative functions) directly from Sample 1 relational norm profiles in a linear mixed regression model. Sample 2 participants were entered as the highest-level grouping variable, with relationship dyad and function-weakening action type then entered as crossed random factors. This variance structure accounts for the fact that for each relationship a judgment was made for every function-weakening action type (i.e., a crossed design). The mean relational norm estimates from Sample 1 were entered alongside both "action likelihood" and "target specificity" as continuous fixed factors for the reasons given above.

The results from this model supported our hypothesis. Relational norms derived from Sample 1 significantly predicted the moral wrongness judgments of Sample 2 participants ($p < .001$, 95% CI [15.63, 16.88]), accounting for 63% of the

variance in mean moral wrongness judgments according to an $R^2$ analysis[51]. Breaking the model down further, we find that target specificity was positively correlated with moral wrongness judgments ($p < .001$, 95% CI [.34, .40]), indicating that the more "off-target" the effect of an action (i.e., in weakening multiple functions) the more harshly that action was judged. Action likelihood was also negatively correlated with moral wrongness judgments ($p < .001$, 95% CI [−.21, −.18]), indicating that rarer actions were judged more harshly, consistent with past research[52]. These results are robust when controlling for demographic factors. For the full regression tables, see Supplement Section 2.5.2.

The "action likelihood" variable serves an additional, theoretically important purpose. As we alluded to previously, it can help to account for the variance in moral judgments that is due to potentially non-moral violations of social-conventional expectations (i.e., deviations from what is socially expected, whether or not the expectation tracks a perceived moral obligation)[50] as opposed to violations of relational norms specifically. By comparing the $R^2$ effect size estimates and Akaike information criterion (AIC) goodness-of-fit scores (i.e., of relational norm versus action likelihood models) we can judge the relative impact of each metric in explaining moral judgments across relationships. We find that, in a model with no information about

| Actions judged to characteristically weaken one or more cooperative functions | | Care Mean (SD) | Hierarchy Mean (SD) | Mating Mean (SD) | Reciprocity Mean (SD) |
|---|---|---|---|---|---|
| 1 | Person A sees Person B crying and walks away from them | -87.9 (15.5) | -15.9 (24.3) | -40.1 (35.0) | -19.0 (32.3) |
| 2 | Person A keeps checking their cellphone while Person B tells a sad personal story | -75.1 (27.8) | -22.1 (29.6) | -36.5 (39.1) | -35 (35.1) |
| 3 | Person A watches passively while Person B carries several heavy boxes up the stairs, even though they could easily help | -73.9 (28.6) | -27.9 (31.3) | -17.4 (29.4) | -29.1 (28.6) |
| 4 | Person A refuses to follow a reasonable order from Person B | -22.1 (29.5) | -89.5 (17.7) | -10.7 (13.1) | -16.9 (24.6) |
| 5 | Person A repeatedly interrupts Person B while they are speaking | -42.2 (28.5) | -71.6 27.7) | -15.9 (20.3) | -50.3 (37.0) |
| 6 | Person A decides to skip a meeting scheduled with Person B without a good excuse | -42.5 (34.7) | -69.8 (30.8) | -21.3 (31.0) | -38.4 (32.4) |
| 7 | Person A refuses to have sex with Person B | -9.3 (17.7) | -10.7 (32.6) | -95.3 (10.7) | 0.1 (2.1) |
| 8 | Person A repeatedly turns down Person B's offer to go on a romantic date | -10.5 (20.2) | -15.3 (19.4) | -77.7 (23.3) | -9.9 (34.3) |
| 9 | Person A invests time and energy in a romantic relationship with someone other than Person B | -26.5 (34.8) | -4.5 (27.6) | -74.0 (28.8) | -31.9 (35.8) |
| 10 | Person A decides not to pay Person B back, hoping Person B won't remember | -37.1 (31.3) | -27.0 (33.9) | -12.4 (17.8) | -85.2 (26.7) |
| 11 | Person A decides not to return Person B's nice favor | -35.5 (29.9) | -9.2 (24.1) | -22.7 (21.6) | -82.0 (21.8) |
| 12 | Person A charges Person B $50 for an item worth $25 | -34.9 (27.9) | -28.1 (31.2) | -13.4 (23.6) | -69.4 (30.7) |

**Fig. 4 Characteristic function-weakening actions.** Heatmap showing mean ratings of judges ($n = 15$) of the extent to which each action characteristically would neglect or violate (weaken) the care, hierarchy, mating, and reciprocity functions, respectively, between any two people (i.e., not assuming the relationship between "Person A" and "Person B" should in fact serve any of those functions). These items were chosen as experimental stimuli from a much larger set by an algorithm using the judges' ratings, where −100 represents the most characteristic function-weakening effect (see Methods). Darker shades represent more extreme ratings. Note: when rating actions on the hierarchy dimension, judges were asked to imagine that Person A was in a *subordinate* role, specifically; when rating actions on the care dimension, judges were asked to imagine that Person A was in a *caregiving* (as opposed to care-seeking) role, specifically. Source data are provided as a Source Data file.

relational norms, action likelihood alone does significantly predict moral wrongness judgments in the absence of other predictors ($p < .001$). However, this model explains much less variance, with a poorer goodness-of-fit score (marginal $R^2 = .08$, AIC = 136,496.9) compared to a model based only on relational norms (marginal $R^2 = .30$, AIC = 130,804). Moreover, the beta value for relational norms (16.26) is more than 80 times larger than that for the action likelihood ratings (−.20) when both are included in the same model (see Supplementary Table 11f in Supplement Section 2.5.2.). This shows that relational norms explain moral wrongness judgments in this study far better than do merely conventional norms regarding what is socially expected.

Having confirmed that relational norms predict between-relationship variation in moral judgments, over and above mere uncommonness or unexpectedness of behavior, we sought to further explore the nature of this predictive relationship. Specifically, we sought to predict the distance between each pair of relationships in moral judgment space (based on Sample 2 patterns of moral judgment) from their corresponding distances in four-dimensional relational norm space (from Sample 1). To do this, we relied on the same K-S distance approach as described above, comparing the moral judgment distributions for each type of function violation for each possible pair of relationships, and averaging across functions to produce an overall moral judgment dissimilarity score for each relationship pair. We then computed a

Spearman's correlation between these moral judgment dissimilarity values and the previously computed relational norm dissimilarity values, hypothesizing that the average K-S distance between every pair of relationships in relational norm space would predict the corresponding K-S distance between the same pairs of relationships in moral judgment space. As can be seen in Fig. 6, this hypothesis was confirmed ($r = .43$, $p = .003$). Looking at the same K-S distances, but on a function-by-function basis (see Supplement Section 2.5.3. for the corresponding scatter-plots), we find that the positive correlation between relational norm and moral judgment K-S distances holds for care ($r = .50$, $p < .001$), mating ($r = .69$, $p < .001$), and hierarchy ($r = .29$, $p = .05$), but not for reciprocity ($r = -.10$, $p = .49$). We will return to this unexpected result for reciprocity in the general discussion.

**Relational norms explain more variance in moral wrongness judgments than alternative models.** Prior work has sought to predict relational variance in moral judgments from factors such as genetic relatedness[53], social closeness[54], and degree of interdependence[55] of the interaction partners. How do these alternative predictors compare to relational norms in terms of explaining variance in relationally situated moral judgments?

To address this question, we asked a third sample of participants (online U.S. convenience sample; final $n = 85$) to

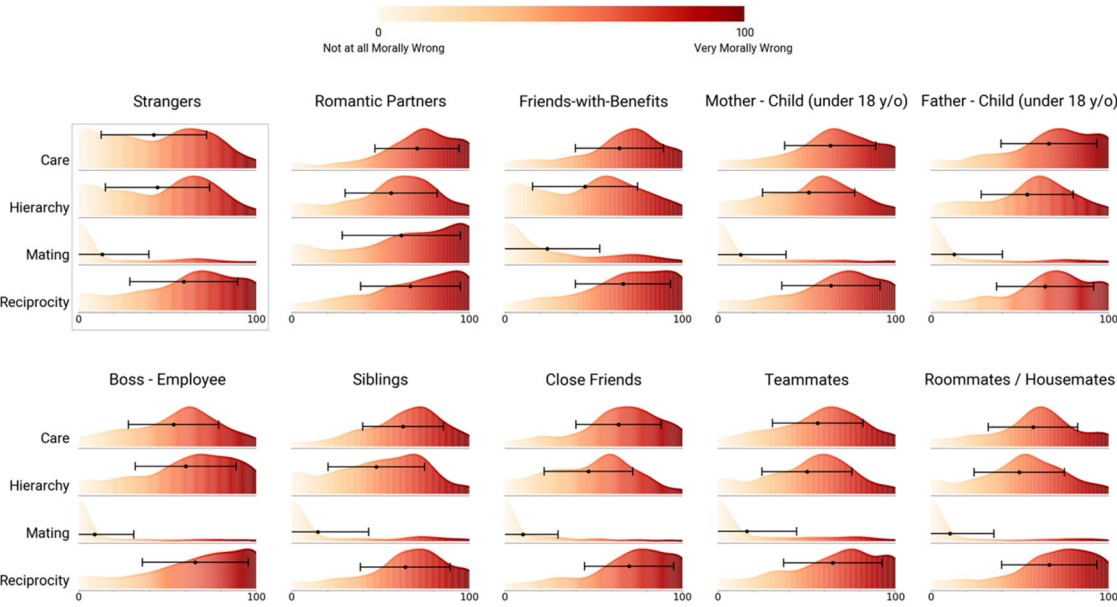

**Fig. 5 Moral wrongness judgments.** Sample 2 moral wrongness judgments for cooperative function violations in different relationships: kernel density plot of wrongness judgments (0 = not at all morally wrong, 100 = very morally wrong) concerning characteristic function-weakening actions for each of four dyadic cooperative functions across 10 relationships. Dot represents the mean, with 95% confidence intervals. Height of the curve represents density (see Fig. 1 for explanation). This experiment was conducted once, with all data shown here. Note that actions which characteristically weaken the mating function (e.g., refusing to have sex with someone) were judged closer to "not at all wrong" than "very wrong" for all dyads apart from the romantic partner relationship. Otherwise, the relative lack of visually dramatic differences in the shape of the moral wrongness judgment distributions between relationships likely can be explained by the mild or "everyday" nature of the function-weakening actions employed in this study (see Fig. 4). Such actions were deliberately chosen to contrast with the more extreme, unusual, or bizarre actions often studied in moral psychology; thus, the ability of our model to predict even subtle variance in moral wrongness judgments between relationships for common, non-extreme actions (see analysis below) can be seen as a strength. Source data are provided as a Source Data file.

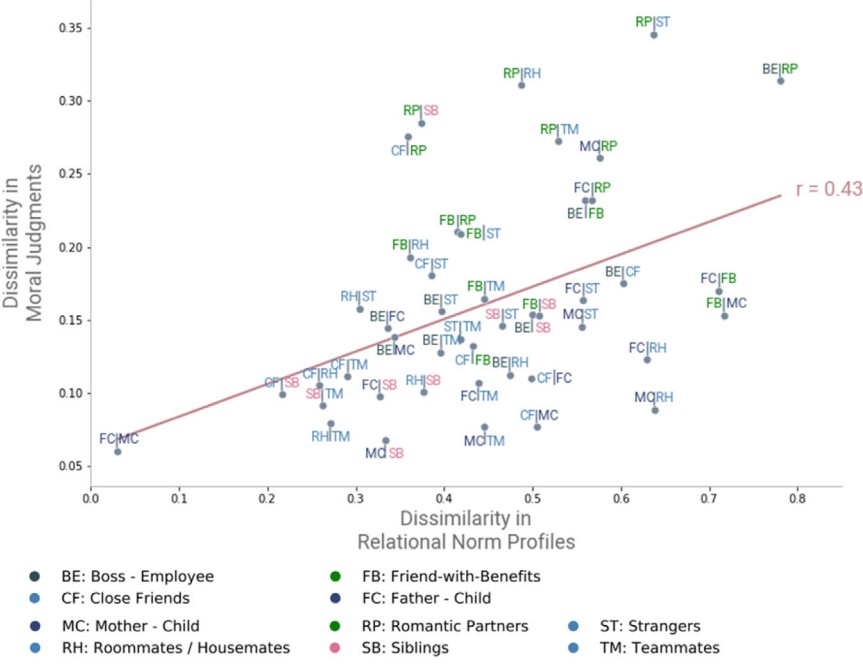

**Fig. 6 Correlation between dyad dissimilarity in relational norms and dissimilarity in moral judgments.** Scatterplot showing the predicted correlation in K-S distance between each pair of relationship dyads in relational norm space (x-axis) and the K-S distance between those same dyads in moral judgment space (y-axis). Spearman's r = .43, p = .003. Note that the color of each relationship reflects the cluster in which it is located from Fig. 3. Source data are provided as a Source Data file.

rate the extent to which a well-functioning instance of each of the 10 distinctive relationship dyads would be characterized by social closeness and interdependence. More specifically, for social closeness, we asked how much the partners would "deeply understand each other," "accept and validate each other's natures," and "strive to care for and promote each other's overall well-being" (taking the mean of these three items). For interdependence, we asked how "frequently," "strongly," and in "how many ways" each partner would affect the other's thoughts, feelings, and behaviors across different situations (ditto). Genetic relatedness for each relationship was determined objectively.

We then entered genetic relatedness, social closeness, and interdependence ratings as predictors in separate linear mixed models similar to those described previously, regressing moral judgments on relational norms. We found no relationship between mean moral wrongness judgments and any of social closeness ($p = .11$, 95% CI [−.04, .39]), interdependence ($p = .14$, 95% CI [−.05, .38]), or genetic relatedness ($p = .78$, 95% CI [−12.41, 9.27]). By contrast, relational norms remained significantly predictive of moral wrongness judgments ($p < .001$, 95% CI [.10, .15]), even controlling for the other factors. In addition, measures of model fit suggest that the relational norms model (marginal $R^2 = .69$, AIC = 841.36) performed substantially better than any of the alternative models: social closeness (marginal $R^2 = .44$, AIC = 908.04), interdependence (marginal $R^2 = .44$, AIC = 908.30), and genetic relatedness (marginal $R^2 = .44$, AIC = 910.33). See Supplement Section 3.4. for the full regression tables.

## Discussion

Several scholars have stressed the importance of taking relational context into account in understanding our moral psychology. Yet the way that relational context has so far been conceptualized so far has suffered from certain limitations. Most commonly, relational context has been understood to vary in a one-dimensional way: for example, in terms of the genetic relatedness of the interaction partners[35], or their social closeness or interdependence[36]. A more promising approach, we think, is to conceptualize relationships in terms of the distinctive pattern of cooperative functions they are normatively expected to rely upon for coordinating behavior in a given society[14,28,39]. Although a number of authors have proposed various taxonomies of cooperative functions[16,37,38] that overlap theoretically with the set employed here[39], it has remained unclear how these functions actually are embedded in different types of relationships. Consequently, we undertook to measure relationship-specific patterns of prescribed cooperative functions (i.e., relational norms) in a U.S. cultural context and to demonstrate how these relational norms predict relationship-specific moral judgments.

To do this, we first measured prescribed cooperative functions for a large set of common dyadic relationships, yielding four-dimensional relational norm profiles for each relationship. Quantifying the distinctiveness of these relationships in terms of their relational norms revealed several distinct clusters of relationship types spanning the domains of care, hierarchy, mating, and reciprocity. Consistent with our predictions, such relational norms predicted out-of-sample moral wrongness judgments in relational context, and explained more relational variance in such moral judgments than genetic relatedness, social closeness, or interdependence of relationship partners. This suggests that moral wrongness judgments of actions within a given relationship are guided by the extent to which the actions violate or neglect prescribed cooperative functions for that relationship. Moreover, relationships with more similar relational norms showed more similar patterns of moral judgments (Fig. 6). These findings reveal

a robust underlying structure of expected relational obligations which shape our moral judgments.

Lewin[56] famously argued that behavior is a product of the person and the situation. In a similar spirit, our data confirm that judgments of moral behavior cannot be understood solely with reference to a given act or actor, but rather, must be interpreted in light of the situation, which, in this case, comprises the type of relationship existing between two individuals. Relationships in a given society can be characterized by distinctive profiles of cooperative norms. They will, therefore, typically be one of the most important situational factors in terms of explanatory power in this domain[57]. Although relationship theorists have, for decades, worked to characterize the structural elements of various close relationships[58] and have sometimes categorized relationships in terms of cooperative functions necessary for human thriving[39,59], here we systematically described lay perceptions of the ideal functional make-up of a wide range of relationships as identified by ordinary language. Moreover, we were able to use this information to make accurate out-of-sample predictions of moral wrongness judgments concerning various actions. We hope that our approach will inspire further research in this vein, both theoretical and empirical, at the interface of relationship science and moral psychology. Ideally, such research will help to integrate and enrich work in both domains, which has so far remained largely separate.

From a theoretical perspective, one aspect of our current account that requires further attention is the reciprocity function. In contrast with the other three functions considered, relationship-specific prescriptions for reciprocity did not significantly predict moral judgments for reciprocity violations. Why might this be so? One possibility is that the model we tested did not distinguish between two different types of reciprocity. In some relationships, such as those between strangers, acquaintances, or individuals doing business with one another[24], each party tracks the specific benefits contributed to, and received from, the other[60]. In these relationships, reciprocity thus takes a tit-for-tat form in which benefits are offered and accepted on a highly contingent basis. This type of reciprocity is transactional, in that resources are provided, not in response to a real or perceived need on the part of the other, but rather, in response to the past or expected future provision of a similarly valued resource from the cooperation partner. In this, it relies on an explicit accounting of who owes what to whom, and is thus characteristic of so-called "exchange" relationships[59].

In other relationships, by contrast, such as those between friends, family members, or romantic partners – so-called "communal" relationships – reciprocity takes a different form: that of mutually expected responsiveness to one another's needs. In this form of reciprocity, each party tracks the other's needs (rather than specific benefits provided)[60] and strives to meet these needs to the best of their respective abilities, in proportion to the degree of responsibility each has assumed for the other's welfare[59]. Future work on moral judgments in relational context should distinguish between these two types of reciprocity: that is, mutual care-based reciprocity in communal relationships (when both partners have similar needs and abilities) and tit-for-tat reciprocity between "transactional" cooperation partners who have equal standing or claim on a resource.

A further limitation of the current studies is that they only concern moral wrongness judgments, based on actions that weaken one or more of the expected relational functions. What about judgments of moral rightness, goodness, or praiseworthiness as these relate to actions which strengthen one or more of the functions[61]? Will people be judged positively for "merely" meeting functional expectations, as when a parent-child

relationship fulfills the care function, or will such judgments be reserved for so-called supererogatory behaviors, going above and beyond the call of duty[62]? Either way, we expect that praise-worthiness judgments for a given action will depend, among other things, on the relational context (functionally understood).

Much of the prior literature in moral psychology has focused on judgments of strangers involved in moral dilemmas that pit distinct ethical principles against one another: for example, a utilitarian imperative to maximize welfare, versus a deontological rule that forbids using individuals as a mere means to an end[5]. A key tenet of utilitarianism is that welfare should be maximized impartially, rather than prioritizing the well-being of close friends or family members (for example) over distant strangers[63]. Descriptive research on moral dilemmas shows that many people are not in fact impartial in this sense[64], consistent with our observations here that people have different cooperative expectations for different relationships, leading in turn to different moral judgments depending on relational context. One intriguing possibility is that individuals who more strongly endorse impartial beneficence will have more uniform prescriptions for cooperative functions across relationships, leading to more uniform moral judgments across relational contexts. This perspective also suggests possible antecedents of impartial beneficence. Because care is normative in close relationships (for example with family, friends, and romantic partners), caring for partners in these relationships does not typically elicit special approbation. Perhaps those who find a sense of purpose or belonging not in tending to close relationships, but in widely being admired[65], tend to "distribute" care across a broader set of relationships (thus showing relatively impartial beneficence).

We note that the generalizability of our findings may be limited in several ways. First, apart from relational role and gender (for mothers and fathers), we did not consider the possible impact of such target characteristics as race, religion, politics, age, and social class on moral judgments. Each variable itself may impact moral judgments[1,66] and interact with relational context in systematic ways. Second, again apart from gender, we did not comprehensively evaluate how observer (i.e., participant) characteristics along those same demographic lines shape moral judgments, thereby impacting the correspondence between relational norms and moral judgments in relational context. Other individual differences among participants, for example in their relative tendency to engage in different styles of moral reasoning[64] will be important to assess in future research. We see our work as a starting point that may launch further investigations into how both target and observer relational qualities interact with each other and with other kinds of characteristics in shaping moral cognition.

Because we studied participants in the U.S., it also will be important to investigate whether our results generalize across different cultures[67]. Although we expect that humans in all cultures form (or stand in) relationships which rely on one or more of the underlying cooperative functions we have highlighted, the patterning of relational norms likely will vary by culture. Indeed, long-standing programs of research have documented such differences using alternative theoretical frameworks. Hindu research participants from the city of Mysore in southern India, for instance, expected care from a broader array of people—from parents, friends, and even strangers—than did research participants from the city of New Haven in the United States[68]. The same difference applied to reciprocity[69]. In another study, American wives felt that husbands should care more for them than for their mothers whereas the reverse held true for Egyptian wives[70]. Future studies might also compare how "tight" (that is, lacking in variance across situations) relational norms are in each culture[71].

Our primary goal for this research has been simple: to investigate how relational context—in particular, the functional cooperative norms that prescriptively govern dyadic interactions of various kinds—shapes moral judgments. A secondary goal has been to push researchers studying human moral psychology to look at behaviors and associated judgments that are more characteristic of people's day-to-day lives than heretofore has been the case. Much remains to be done, including more precise and sophisticated analyses of which cooperative functions apply to which relationships, how these functions relate to one another, and how they can be used to predict praiseworthiness judgments (not just judgments of moral wrongness as we have undertaken here). As we and others pursue work that places the study of morality in both geopolitical and relational cultural context, we anticipate the emergence of a more nuanced literature on human morality that becomes better integrated with broader and long-standing programs of research on relationships and prosocial behavior.

## Methods

All studies were reviewed and approved by the Yale University Institutional Review Board (protocol #20000022385); informed consent was obtained from participants in each instance prior to data collection. We have posted all study materials, pre-registration forms, raw data, and analysis code on the Open Science Framework (https://osf.io/zxjt6/). For complete study descriptions and supplementary findings, see the Supplement.

*Stage 1.* For Stage 1, the design, measures, sampling plan, and exclusion criteria were pre-registered at aspredicted.org (#26400). We used an online polling software (https://www.nbrii.com/our-process/sample-size-calculator/) to determine that at least 385 participants would be needed to obtain population estimates of cooperative functional expectations with a 5% margin of error and 95% confidence level. Anticipating participant exclusions, we over-sampled by about 15% and aimed to recruit 450 U.S. participants via the Prolific Academic platform (Prolific); 493 ultimately took the survey, each of whom was paid at a rate of $7.25 per hour. Seventy (70) participants were excluded based on the pre-registered exclusion criteria, leaving us with a final sample of 423 participants ("Sample 1") who completed an online survey. Participants were given descriptions and definitions of all five cooperative functions adapted from Bugental (57): care, coalition, hierarchy, mating, and reciprocity (see Supplement Section 1.2.1. for the full descriptions). To ensure that participants were thinking of the functions in the way we intended, participants were not able to advance to the main part of the study before passing multiple comprehension checks.

We then asked participants to indicate how much each of 20 common relationships ideally should serve each of the five cooperative functions, specifying: "if this kind of relationship was the best possible relationship of its kind it could be [i.e., according to general societal standards], how much should it serve each of those 5 relationship functions?" Participants rated each relationship type in random order. For each relationship type, we included a brief description (see Supplement Section 1.2.2. for the descriptions). Then, for each combination of relationship and function, participants rated how much the relationship ideally should serve the given function on a sliding scale from −100 (definitely should not serve) to +100 (definitely should serve). Since every participant assessed all five functions for all 20 relationships, we obtained 100 data points per participant. Finally, we collected a battery of demographic measures (described next) as well as exploratory measures for future studies not included here.

In analyzing the demographic information, we first excluded the coalition ratings for the reasons described in the main text. We then used a linear mixed model to regress prescribed cooperative function scores on participant gender (female, male; 4 participants who marked 'other' were excluded) for each of the four remaining functions. Reported annual income ("low" = $35 K or less, "high" = more than $35 K; split based on U.S. median income), religiosity ("high" versus "low" based on a mean split), and both social and economic political ideology (ditto) were entered into the model as categorical covariates. Full model summaries are in Supplement Section 1.4.5.

*Stage 2.* For Stage 2, the hypothesis, design, measures, sampling plan, exclusion criteria, and analysis approach were pre-registered at aspredicted.org (#31592). Two main steps were involved: first, selection of a subset of relationships from Stage 1 plus the generation of "action items" to be rated for subsequent use; and second, the actual study, collecting ratings from a new sample ("Sample 2"). As before, the Sample 1 coalition data were excluded.

Using the (remaining) Sample 1 cooperative function scores for all 20 relationships, we performed an analysis that is conceptually similar to a representational similarity analysis (RSA), except that it relies on the Kolmogorov-Smirnov (K-S) distance statistic rather than distribution means. The goal of this analysis was to identify relationships with relatively dissimilar relational norm profiles, so that 10 of the least functionally redundant relationships could be used in the current

Stage. For the RSA-like analysis, each relationship was compared to every other relationship on the dimensions of care, hierarchy, mating, and reciprocity. The mean of the four corresponding K-S distance statistics was used for this comparison. Next, we ranked each pair of relationships by its mean K-S distance, from least to most distant (that is, from most functionally redundant to least functionally redundant). We then dropped the relationship from each pair that had the lowest mean K-S distance from all other relationships in the set. Note: for theoretical reasons (i.e., to allow gender comparisons) we decided in advance to retain both the father-child and mother-child relationships in case they faced off. The final set of relationships identified by this procedure is shown in Fig. 2 of the main text.

We then created a set of 86 "action statements" describing behaviors that would plausibly neglect or violate (i.e., weaken) specific cooperative functions based on their underlying logic (i.e., how each function solves its corresponding coordination problem). To determine the extent to which certain actions would characteristically weaken (or strengthen) each of the four dyadic cooperative functions, we had 15 trained judges rate all 86 action items in our set. These judges were recruited among lab members and colleagues and were given extensive training, either in-person or using an online video conferencing platform, to ensure high quality ratings. They were instructed to consider only the functional implications of each action, setting any potential moral considerations strictly aside.

Ratings were obtained via an online survey. The survey included the same formal descriptions of cooperative functions used in Stage 1. Following multiple comprehension checks, the judges were shown the 86 action statements, in random order, in the format "Person A does X to Person B." For each action and function combination, they made their judgment on a sliding scale ranging from "Would characteristically weaken [the function]" ($-100$) through "It depends/Would neither weaken nor strengthen" (0) to "Would characteristically strengthen" ($+100$).

Next, we created an algorithm to select 12 action items that were rated among the most characteristic in weakening each of the four cooperative functions (three statements per function; 'weaken' set). First, for each function, the algorithm ranked the actions, in ascending order, by their mean weakening "characteristicness" rating and randomly selected 3 out of the seven most characteristic actions. Second, it computed the mean rating across the three selected actions, yielding one mean score per function. Third, the algorithm computed the standard deviation of the four function-specific means generated in the previous step. Finally, steps one to three were repeated 10,000 times to find the combination of three action statements that yielded the lowest standard deviation of scores across functions. The second iteration of the algorithm was subjected to two further constraints so that we could ensure consistency with potential function-strengthening items planned for testing in future studies ('strengthen' set). The first constraint was that the minimum mean score in the 'weaken' set could not be lower than the minimum mean score in the 'strengthen' set. The second constraint was that the average of the final 'weaken' scores could not be more than one point lower than the average of the final 'strengthen' scores. So that future studies can be straightforwardly compared with the present study, we selected items for the 'weaken' set so that they would weaken the cooperative functions to a similar degree as future 'strengthen' items would strengthen the functions. In other words, we wanted to make sure that we identified a set of 'weaken' items that were not more extreme (in the 'weaken' direction) than future 'strengthen' items would be (in the 'strengthen' direction). This process resulted in a final set of 12 function-weakening action statements, with three per function, as shown in Fig. 4.

Proceeding to the second main part of Stage 2, a set of naive/lay participants (Sample 2) was recruited, this time on Amazon's Mechanical Turk (MTurk). To power for the same confidence and margin of error as in Sample 1, but this time with a between-subjects design, it was determined that we would need ratings from 1,551 participants (see Supplement Section 2.2. for the full rationale). Based on the Sample 1 exclusion rate, we over-recruited by about 10% and thus aimed to recruit 1,706 participants; 1,822 ultimately filled out at least part of the survey (not all finished), each of whom was paid $1.00. Five hundred and two (502) participants were excluded based on pre-registered exclusion criteria, leaving us with a final sample of 1,320 participants. As in Stage 1, they were shown brief descriptions of their assigned relationship. They were told that they would be asked to rate the moral wrongness of various actions within the relationship. To orient them to the rating scale, we clarified that none of the actions they would see would be extreme (e.g., murder), but rather would all be actions that might plausibly occur within the course of day-to-day life.

After passing several attention and comprehension checks, participants were shown, in random order, all 12 action items, tailored to their assigned relationship. For instance, if they were assigned the romantic partner relationship, one of their items was: "Imagine that someone keeps checking their cell phone while their romantic partner tells a sad personal story. How morally wrong would that be, if at all?" Responses were recorded on a sliding scale from "Not at all morally wrong" (0) to "Very morally wrong" (100). Finally, we collected data about how likely or unlikely it was that each of the rated actions would happen in real life, and administered the same battery of demographic measures as were used in Stage 1.

For the K-S distance analysis reported in the main text, please note that the functional ratings from Sample 1 were first z-scored to each Sample 1 participant in order to account for individual differences in scale use; for the moral wrongness ratings, no such z-scoring was performed because each Sample 2 participant made only 12 ratings (on account of the between-subjects design). For the linear mixed regression model reported in the main text, although the moral wrongness variable

was not normally distributed, Q-Q plots indicated that this did not violate the normality assumption of the model. See Supplement Section 2.5.1. for details.

*Stage 3.* For Stage 3, we powered to have as many observations per distribution as were obtained in Stage 1. Given design differences between the studies, we determined that we would need ratings from 150 participants for the current study (see Supplement Section 3.1. for the full rationale). This is the number we recruited on MTurk; ultimately, 149 participants completed the survey, each of whom was paid $1.00 for their time. Sixty-four (64) participants were excluded based on the predetermined exclusion criteria, leaving us with a final sample of 85 participants. Participants were shown the same descriptions of relationships used in Stage 2 and asked to rate them along three dimensions each of social closeness and inter-dependence (see Supplement Section 3.2.1. for the precise wording). Responses—regarding the extent to which a well-functioning instance of each relationship would be characterized by each dimension of both constructs—were recorded on a sliding scale from 0 to 100, labelled appropriately for each dimension. Similar demographic measures to those used in the previous studies were administered. Please note that, given the unexpectedly large proportion of excluded participants in this study, we performed a sensitivity/robustness analysis with no exclusions (see Supplement Section 3.4.2. for details), and the results remain substantively the same.

**Reporting summary**. Further information on research design is available in the Nature Research Reporting Summary linked to this article.

## Data availability
All original data (anonymized) and study materials are available on the Open Science Framework at https://osf.io/zxjt6/. Source data are provided with this paper.

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

## Acknowledgements

Thank you to Billy Brady, Elena Khusainova, Henry Glick, Kevin Anderson, Clara Colombatto, and Hongbo Yu for statistical advice and coding assistance on this project. Thank you to Rachel Calcott, Aden Goolsbee, Nell Mermin-Bunnell, Yuri Munir, Lillian Yuan, Vivan Fung, Vanessa Copeland, Heeral McGhee, Alan Presburger, Samar Allibhoy, Elena DeBre, Isobel Munday, Gargi Singh, Stephanie Brown, Ryan Cox, Brian Bink, Qihe Sun, and Daniel Do, for help in drafting stimuli or serving as judges to rate items used in the study. Finally, thank you to the members of the Crockett, Clark/Bargh, Bloom, and Knobe labs at Yale University for feedback on this research and/or comments on earlier drafts of this manuscript.

## Author contributions

Designed research: B.D.E., J.T.M., M.S.C., M.J.C. Performed research: B.D.E., J.T.M. Analyzed data: K.L.M., B.D.E. Wrote the paper: B.D.E., J.T.M., M.S.C., M.J.C., K.L.M.

## Competing interests

The authors declare no competing interests.
