## [Peer Review File · Nature Communications]

REVIEWER COMMENTS

Reviewer #1 (Remarks to the Author):

I enjoyed reading this paper and think it's a great contribution to moral psychology. Several researchers have previously proposed accounts of morality that center on relationships and cooperative norms, but as the authors state, the current state of the field is light on data supporting those theories. The authors present sound, rigorous, and reproducible data demonstrating how the cooperative norms that govern dyadic relationships powerfully shape moral judgments. This is a straightforward point that is important to make.

My main comment on the paper centers around the question of what theoretical contribution the paper makes. In the abstract, the authors suggest they "provide new theory." The introduction suggests that prior work is not "theoretically coherent" or missing a "unifying theory." And while the authors acknowledge some previous research outlying similar theories, it is unclear when they say "we develop our theory" what theory they are developing. The paper is strongest when it focuses on its empirical support for the general theory that cooperative norms matter for moral judgment. It is less successful in making a theoretical contribution beyond what has been suggested in the literature previously.

If the authors do want to make a theoretical contribution, the paper would need to do much more to explain precisely what it is and how it differs from the other accounts. For example, it's unclear if the 4 dyadic functions they focus on are meant to represent a more coherent or unifying framework than other ways of characterizing coordination problems, such as Curry et al.'s 7-factor MAC theory. My reading is that the 4 functions focused on here are not meant to be all encompassing, but rather a strong starting point and proof of concept for how these functions predict moral judgment. I think the value of the paper is strong enough without devoting the space necessary to make a stronger novel theoretical claim.

A couple methodological points:

- The logic behind the two constraints on the algorithm used to select the action items is unclear. What is the weaken set vs. the strengthen set and weaken score vs. strengthen score? Weren't the items rated on a single bipolar scale from "would characteristically weaken" to "would characteristically strengthen?" And why could the weaken scores not be more than one point lower than the strengthen scores?

- On pg. 28, the phrasing for the relationship function as actually used in the questions should be included. Table S3 has the full descriptions, but the actual measures use a condensed wording (e.g., "the function of giving or receiving unconditional support (care function)") that is likely more impactful on people's ratings than the longer descriptions presented earlier.

Reviewer #2 (Remarks to the Author):

Review of NCOMMS-20-36556 “How social relationships shape moral judgment”

Let me begin by noting that I am not a psychologist, let alone a moral psychologist. Thus my comments should be viewed as coming from someone in a neighboring discipline (sociology) who does some work on related issues. My overall assessment of the paper is that it addresses an important issue in a clearly written and methodologically clean and sophisticated way. Generally, I like the paper and think it makes an important contribution. But I do have a few related concerns (and suggestions) that I hope the authors will seriously consider.

My main concern is that the paper, as currently framed, falls a bit flat and comes off as less interesting than perhaps it could. The introduction made me eager to find out what the results might be. But when I got to the predictions and results, I could not help thinking that it would be hard to expect anything other than what the authors are predicting and finding. This is where my read as an outsider may be useful to the authors. I am not familiar with the theory on which the paper builds and thus what may seem particularly novel from the perspective of that theory (I don't know) comes off as somewhat obvious from my standpoint. By midway through the results, I was wondering what the theoretical stakes are.

But there are theoretical stakes, as the authors show when they compare their model with other theories (genetic relatedness, theories of interdependence, etc.). So the reader learns that there are set of handful of theories that fare worse in predicting these outcomes than the theory tested here. But this comes much later (too late in my view), is not anticipated in the introduction, revisited in the discussion, and isn't really developed much. As a result, it reads as a relatively unimportant aside. I would encourage the authors to anticipate and feature these rival theories to i) show clearly early on that other theories make predictions for these data, and ii) more convincingly show that the theory the authors are presenting makes unique/non-obvious/interesting predictions.

A second, related, concern is also a “so what” style question (sorry!). The hook of the paper is all about prior work in moral psychology focusing on “raceless, genderless, strangers.” This paper also focuses on raceless, genderless others. That leaves strangers. But at most junctures in the data, strangers, it felt to me that we got evidence that “stranger” is a nice “representative” profile to study – they are “less functionally polarized” (as you note) but that could be framed as a plus, not a con. Similarly, the figures (e.g., Figure 4) show a lot of similarity across relationship types. None of this is to devalue the aims of the paper. But I do think it means that the authors should slightly recast the introduction (or at least more critically revisit that introduction later in the paper) to address the fact that while moral psychologists have recently castigated one another for looking only at (raceless, genderless) strangers, the authors have taken them up on a call look at relational context and while there is (unsurprisingly) some variation there, it is not as large as one might expect. Ideally, a discussion of tradeoffs of what advances in theoretical and substantive knowledge we get from increasingly complicated models and

experimental designs would follow that. (Or perhaps the authors disagreed with me here and can show that the relationship profiles really show a large degree of qualitative differences. That would also be useful, as I did not get this sense from the current draft.)

Reviewer #3 (Remarks to the Author):

This manuscript contains an intriguing study about moral judgment in the context of specific social relationships. I enjoyed reading this manuscript. It's well-written and theoretically grounded. There is a clear underlying logic for the key prediction that participants would perceive actions as morally wrong to the extent that they disrupt the functional expectations of the relationship. Being that this is my research area (morality and relationships), I am eager to see more of this work published. I have little doubt that this study will be of interest to readers of the journal. That being said, I have some concerns that the authors should address before this can be publishable.

I don't think the authors have adequately clarified what exactly functional expectations have to do with morals and ethics compared to sociocultural norms. Research has shown that adult populations (and even small children) can tell the difference between moral norm violations (e.g., stealing) and social norm violations (e.g., standing on the wrong side of an escalator), so this is something that research participants in all demographics would likely be aware of. It seems that the authors are suggesting that the functional expectations of a relationship (X) are rooted in sociocultural norms, which then predict patterns of moral judgment (Y). In the Discussion, the authors stated that, "moral judgments of actions within a given relationship are guided by the extent to which the actions violate cooperative norms for that relationship." This logic of this XY association is intuitive. However, I am curious whether all types of expectations and norms (X) fall in the sociocultural norm category, or if some of them are infused with moral considerations to begin with. Put another way, I am curious if the authors can tease apart the specific moral and normative components within each functional expectation. Perhaps both types end up predicting moral judgment in unique ways (which would be very interesting), although my hunch is that the moral expectations are much stronger predictors of moral judgment than the sociocultural ones. Both friendship and teammate relationships may be hurt by Reciprocity violations, but perhaps friend relationships are hurt because the expectations are moral, while teammate relationships are hurt because the expectations are sociocultural.

In the authors' work, which functional expectations are sociocultural, and which are moral? My guess would be that certain elements of Care, Reciprocity, and Hierarchy are moral to begin with, as they map onto moral dimensions in other lines of research (e.g., Moral Foundations Theory). But is Mating moral? Is "turning down someone for a date" a moral violation or is it a sociocultural violation, in the context of romantic partners? Similarly, Reciprocity expectations like "deciding not to return a nice favor" may be immoral in the context of close friendships, but amoral in the context of housemates or neighbors. Collapsing across social and moral norms poses a problem conceptually, but it also poses a problem analytically when it appears based on the Sample 2 density plot data that there is little difference

between siblings and teammates, housemates, bosses, or strangers.

This distinction may help the authors clarify the null association they reported for the Reciprocity function. The authors suggested that this could be explained by the difference between exchange/transactional and communal relationship norms, which is reasonable, but perhaps other explanations are also possible.

The authors are absolutely right to note that prior research about moral judgment toward “faceless strangers” does not apply to most of the morally-relevant interactions people have with others in their social networks. However, that prior work has yielded many insights that the authors may benefit from incorporating into their analyses, or at least some suggestions future work. For instance, some individuals are more prone to perceiving something as moral because it maximizes welfare for the most people even if minor harms are caused (utilitarianism) or because it follows prescriptive rules (deontology) or because of downstream aftereffects (consequentialism). How do these inform moral judgments in social contexts? Do people rate close friends/family members high in Care or Reciprocity because of deontological norms or because of utilitarianism? Or do these have nothing to do with each other?

Overall, provided that the authors can clarify/distinguish between the moral and sociocultural components of functional expectations (which may require further analytical information), and address the ways in which moral principles would guide moral judgment across relationship contexts, I think this paper is of publishable quality.

REVIEWER #1 (R1)

I enjoyed reading this paper and think it's a great contribution to moral psychology. Several researchers have previously proposed accounts of morality that center on relationships and cooperative norms, but as the authors state, the current state of the field is light on data supporting those theories. The authors present sound, rigorous, and reproducible data demonstrating how the cooperative norms that govern dyadic relationships powerfully shape moral judgments. This is a straightforward point that is important to make.

We are pleased to receive such positive feedback, and sincerely thank this reviewer for taking the time to provide such thoughtful comments on our manuscript, which we address below. This feedback, and that of the other reviewers, has helped us to substantially improve our paper.

R1.1. Clarifying theoretical vs. empirical contributions of the work

My main comment on the paper centers around the question of what theoretical contribution the paper makes. In the abstract, the authors suggest they “provide new theory.” The introduction suggests that prior work is not “theoretically coherent” or missing a “unifying theory.” And while the authors acknowledge some previous research outlying similar theories, it is unclear when they say “we develop our theory” what theory they are developing. The paper is strongest when it focuses on its empirical support for the general theory that cooperative norms matter for moral judgment. It is less successful in making a theoretical contribution beyond what has been suggested in the literature previously. If the authors do want to make a theoretical contribution, the paper would need to do much more to explain precisely what it is and how it differs from the other accounts. For example, it's unclear if the 4 dyadic functions they focus on are meant to represent a more coherent or unifying framework than other ways of characterizing coordination problems, such as Curry et al.'s 7-factor MAC theory. My reading is that the 4 functions focused on here are not meant to be all encompassing, but rather a strong starting point and proof of concept for how these functions predict moral judgment. I think the value of the paper is strong enough without devoting the space necessary to make a stronger novel theoretical claim.

Thank you for these comments. You are absolutely right that the particular dyadic functions we focus on, adapted from Bugental's model, are not themselves at the heart of our theoretical contribution. As we now clarify in the introduction, a number of scholars have proposed sets of cooperative functions that overlap theoretically with those proposed by Bugental. For example, the 'reciprocity' function in Bugental's model shares certain features with 'social exchange/reciprocity' and 'division/fairness' in Curry et al.'s MAC model, 'equality matching' in Fiske's Relational Models Theory (RMT), and 'fairness/proportionality' in Haidt's Moral Foundations Theory (MFT). Similarly, the 'hierarchy' function in Bugental's model overlaps to some extent with 'respect' in the MAC model, 'authority ranking' in RMT, and 'authority/respect' in MFT, and so on.

To address the concern about potentially overstating our theoretical contribution, we have removed all claims of theoretical novelty from the abstract and paper. In addition, to drive home the point that the specific functions we have adapted from Bugental are not definitive or all-encompassing, as you correctly state, but are rather among a number of functions we might have employed, we write (p. 3 of the revised manuscript):

In contrast to genetic relatedness, which can be determined objectively, and the constructs of social closeness and interdependence, both of which have been carefully defined within the relationship science literature, there is no agreed-upon set of cooperative functions prescribed for different social relationships to solve characteristic coordination problems. Recognizing both the theoretical overlap and diversity among the various existing taxonomies of cooperative functions (1–3), we build on work by Bugental (4).

You are also right that a major impetus behind this research was to show, in a more rigorous, empirically-driven way, how a set of common cooperative functions shape moral judgments in relational context. As a part of this, we began with the assumption that multiple cooperative norms are, themselves, embedded within different relationship types (kin relationships, work relationships, relationships with strangers and so forth); and this does differentiate our approach from most of what has been done previously. MAC theory, for instance, identifies "family" as one basis for morality, but refers to no other specific type of relationship as being one within which cooperative norms are embedded. For us,

it is not enough to say morality can be based on this or that distinct cooperative norm; we go beyond this to say that the distinct *patterns* of cooperative norms embedded within different social relationships shape moral judgments of behavior occurring within those relationships. As we now state in the discussion section of the manuscript (pp. 16-17 of the revision):

Most commonly, relational context has been understood to vary in a one-dimensional way: for example in terms of the genetic relatedness of the interaction partners (5), or their social closeness or interdependence (6). A more promising approach, we think, is to conceptualize relationships in terms of the distinctive cooperative functions they are normatively expected to rely upon for coordinating behavior in a given society (4,7,8). Although a number of authors have proposed various taxonomies of cooperative functions (1–3) that overlap theoretically with the set employed here (4), it has remained unclear how these functions actually are embedded in different types of relationships. Consequently, we undertook to measure relationship-specific patterns of prescribed cooperative functions (i.e., relational norms) in a U.S. cultural context and to demonstrate how these relational norms predict relationship-specific moral judgments.

R1.2. Clarifying the methods for selecting action items

The logic behind the two constraints on the algorithm used to select the action items is unclear. What is the weaken set vs. the strengthen set and weaken score vs. strengthen score? Weren't the items rated on a single bipolar scale from "would characteristically weaken" to "would characteristically strengthen?" And why could the weaken scores not be more than one point lower than the strengthen scores?

Apologies for being unclear. We are planning future studies (beyond the scope of the present paper) to examine positive moral judgments of function-strengthening actions ('strengthen' set) in contrast with negative moral judgments of function-weakening actions ('weaken' set, the focus of the present study). So that future studies can be straightforwardly compared with the present study, we selected the 'weaken' action items so that they would weaken the cooperative functions to a similar degree as future 'strengthen' items would strengthen the cooperative functions. The reason for doing this is because we wanted to make sure that we identified a set of 'weaken' items that were not more extreme (in the 'weaken' direction) than future 'strengthen' items (in the 'strengthen' direction), which would limit the comparisons we could make in future studies. We have now

added this clarification to the methods section of the revised manuscript (on p. 22.)

R1.3. Minor comment: recommendation to include specific wording

On pg. 28, the phrasing for the relationship function as actually used in the questions should be included. Table S3 has the full descriptions, but the actual measures use a condensed wording (e.g., “the function of giving or receiving unconditional support (care function”) that is likely more impactful on people’s ratings than the longer descriptions presented earlier.

Thank you for this recommendation. We have now included both the full and brief/condensed descriptions in the table.

REVIEWER #2 (R2)

My overall assessment of the paper is that it addresses an important issue in a clearly written and methodologically clean and sophisticated way. Generally, I like the paper and think it makes an important contribution. But I do have a few related concerns (and suggestions) that I hope the authors will seriously consider.

We are grateful to the reviewer for their helpful, constructive feedback on the paper. We hope from our comments below, and the associated revisions to the manuscript, the reviewer will see that we have indeed given serious consideration to the concerns raised, and will judge that we have addressed them to their satisfaction.

R2.1. Clarifying the theoretical stakes of the paper

My main concern is that the paper, as currently framed, falls a bit flat and comes off as less interesting than perhaps it could. The introduction made me eager to find out what the results might be. But when I got to the predictions and results, I could not help thinking that it would be hard to

expect anything other than what the authors are predicting and finding. This is where my read as an outsider may be useful to the authors. I am not familiar with the theory on which the paper builds and thus what may seem particularly novel from the perspective of that theory (I don't know) comes off as somewhat obvious from my standpoint. By midway through the results, I was wondering what the theoretical stakes are. But there are theoretical stakes, as the authors show when they compare their model with other theories (genetic relatedness, theories of interdependence, etc.). So the reader learns that there are set of handful of theories that fare worse in predicting these outcomes than the theory tested here. But this comes much later (too late in my view), is not anticipated in the introduction, revisited in the discussion, and isn't really developed much. As a result, it reads as a relatively unimportant aside. I would encourage the authors to anticipate and feature these rival theories to i) show clearly early on that other theories make predictions for these data, and ii) more convincingly show that the theory the authors are presenting makes unique/non-obvious/interesting predictions.

We thank the reviewer for this valuable advice. We agree that the theoretical stakes were not set up as clearly as they should have been. We have now rewritten a substantial portion of the introduction to clarify these stakes in line with the suggestion raised. In particular, we draw attention to the comparison we make *between* different ways of characterizing social relationships in terms of explanatory power in predicting relationally-situated moral judgments. Thus, as we now state explicitly in the introduction, some recent work has indeed tried to predict moral judgments in relational context, but has done so by characterizing relationships *not* in terms of the multiple cooperative functions that each one serves (as we do here), but in terms of a unidimensional construct, such as the genetic relatedness of the interaction partners, their social closeness, or their interdependence. By showing that our multi-dimensional cooperative function approach has much greater predictive power in explaining moral judgements in relational context than the other recent one-dimensional models, we go beyond what has been shown in previous work. As we now state in the manuscript (pp. 2-3 of the revision):

A number of theorists have highlighted relational context as likely to be important for understanding moral judgment and behavior (1,9–13). In line with these developments, there is now a small but growing empirical literature which

explores how moral judgments of particular actions vary across different types of social relationships (8,14–28). How these relationships are theorized depends on the study. For example, one recent study characterized relationships in terms of the *genetic relatedness* of the interaction partners, and showed how varying this factor affects moral judgments about helping behavior (5). Another recent study characterized relationships in terms of the authors' intuitive sense of the *social closeness* and relative *interdependence* of the interaction partners – regardless of genetic relatedness – and tested the influence of these factors on judgments about violations of care (6). Researchers have also sought to predict moral wrongness judgments of actions in relational context from a single *cooperative function* thought to characterize a given relationship (e.g., care for a sibling relationship, hierarchy for a teacher-student relationship, and so on) (8).

These studies demonstrate that moral judgments of one and the same action often differ across different types of relationships, depending on how relationship “type” is understood. What is missing, however, is a systematic, data-driven account of the *multiple* cooperative functions that can characterize any given social relationship (7), and an explicit comparison of how well such cooperative functions predict relationally-situated moral judgments relative to alternative models such as genetic relatedness, social closeness, and interdependence. We aim to fill that gap with the present research.

Then, in the results section of the paper, we now highlight our statistical comparisons of the respective models, creating a separate sub-section to frame this more clearly (see p. 16 of the revised manuscript).

R2.2. Highlighting between-relationship variation in cooperative norms as compared to the ‘stranger’ relationship

A second, related, concern is also a “so what” style question (sorry!). The hook of the paper is all about prior work in moral psychology focusing on “raceless, genderless, strangers.” This paper also focuses on raceless, genderless others. That leaves strangers. At most junctures in the data, strangers, it felt to me that we got evidence that “stranger” is a nice “representative” profile to study – they are “less functionally polarized” (as you note) but that could be framed as a plus, not a con. Similarly, the figures (e.g., Figure 4) show a lot of similarity across relationship types. None of this is to devalue the aims of the paper. But I do think it means that the authors should slightly recast the introduction (or at least more critically revisit that introduction later in the paper) to address the fact that

while moral psychologists have recently castigated one another for looking only at (raceless, genderless) strangers, the authors have taken them up on a call look at relational context and while there is (unsurprisingly) some variation there, it is not as large as one might expect. Ideally, a discussion of tradeoffs of what advances in theoretical and substantive knowledge we get from increasingly complicated models and experimental designs would follow that. (Or perhaps the authors disagreed with me here and can show that the relationship profiles really show a large degree of qualitative differences. That would also be useful, as I did not get this sense from the current draft.)

We appreciate the reviewer's perspective here, and have made a number of edits to the manuscript to address these points. First, we have removed the quote about "raceless, genderless" strangers: it is true that we, ourselves, have not incorporated the study of race into this work nor have we focused much on gender (apart from the mother/father distinction). As the reviewer notes, this leaves "strangers" as one of the relationships we do explore, but we need not imply that this relationship is theoretically unimportant. Consequently, in the opening paragraphs, we now clarify that the stranger-stranger relationship is, in fact, one important type of relationship to study. As such, our aim is not to discount the stranger-stranger relationship, but rather, to expand the range of relationships beyond the usual small set that has been explored in previous studies. As we now state on p. 2 of the revision:

Of course, people often do encounter strangers as they go about their lives, and the interpersonal standing implied by such encounters can be seen as a bare-bones social relationship involving certain minimal obligations: for example, a "duty of easy rescue" in the case of emergencies (29). The copious research on moral judgments in the context of stranger-stranger relationships thus sheds important light on at least one important aspect of our moral psychology.

We also appreciate the reviewer's point that, in a sense, the stranger-stranger relationship could be seen as a 'representative' or perhaps 'default' social relationship, in that it is not, for instance, as functionally polarized as some of the other relationships. To capitalize on this insight, we have now reformatted the figures in the paper to set the 'stranger' relationship off from the other relationships as a kind of 'baseline' against which the various dimensions of qualitative difference in functional expectations for the other relationships can

more clearly be seen. On the next page is a screenshot of part of our new Figure 1, to illustrate (see below).

As we think this revised figure now more clearly shows, there is in fact systematic variation ‘away’ from the stranger default along various dimensions for certain relationships. Starting on the left-hand side of the figure, looking at Care, it can now clearly be seen, for instance, that the Romantic Partner, Child-Parent, and Close Friend relationships are all rather dramatically ‘shifted up’ from the Stranger ‘baseline’ with a much greater expectation of Care. For Hierarchy, the Child-Parent, Teacher-Student, and Boss-Employee relationships are similarly ‘shifted up’ compared to the Stranger relationship in expectations for Hierarchy. For Mating, the Romantic Partner and Friends with Benefits relationships are ‘shifted up’ while the Child-Parent, Teacher-Student, and Boss-Employee relationships are ‘shifted down.’ And finally, for Reciprocity, it can be seen that, in addition to the Romantic Partner relationship, the set of relationships starting with Close Friends (below Boss-Employee) are all ‘shifted up’ from the stranger ‘baseline’ as well.

With respect to the patterns of moral judgments depicted in Figure 4, we agree with you these are not dramatically different between relationships, at least to the naked eye. Accordingly, we have added material to the caption for Figure 4 (see page 13 of the revised manuscript), to provide a potential explanation for this relative lack of obvious between-relationship variance:

... the relative lack of visually dramatic differences in the shape of the moral wrongness judgment distributions between relationships can likely be explained by the mild or 'everyday' nature of the function-weakening actions employed in this study (see Table 2). Such actions were deliberately chosen to contrast with the more extreme, unusual, or bizarre actions often studied in moral psychology; thus the ability of our model to predict even subtle variance in moral wrongness judgments between relationships for common, non-extreme actions (see analysis below) can be seen as a strength.

REVIEWER #3 (R3)

This manuscript contains an intriguing study about moral judgment in the context of specific social relationships. I enjoyed reading this manuscript. It's well-written and theoretically grounded. There is a clear underlying logic for the key prediction that participants would perceive actions as morally wrong to the extent that they disrupt the functional expectations of the relationship. Being that this is my research area (morality and relationships), I am eager to see more of this work published. I have little doubt that this study will be of interest to readers of the journal. That being said, I have some concerns that the authors should address before this can be publishable.

We thank the reviewer for the positive feedback, as well as for raising the important theoretical considerations concerning, among other things, the distinction between social-conventional expectations and moral norms. We have revised the manuscript -- and performed additional data analysis -- to respond to the reviewer's concerns, and we believe that doing so has strengthened and clarified the basis for our contribution. Please see our responses below for more information.

R3.1. Clarifying the distinction between cooperative functional norms and nonmoral sociocultural or conventional norms

I don't think the authors have adequately clarified what exactly functional expectations have to do with morals and ethics compared to sociocultural norms. Research has shown that adult populations (and even small children) can tell the difference between moral norm violations (e.g., stealing) and social norm violations (e.g., standing on the wrong side of an escalator), so this is something that research participants in all demographics would likely be aware of. It seems that the authors are suggesting that the functional expectations of a relationship (X) are rooted in sociocultural norms, which then predict patterns of moral judgment (Y). In the Discussion, the authors stated that, "moral judgments of actions within a given relationship are guided by the extent to which the actions violate cooperative norms for that relationship." This logic of this XY association is intuitive. However, I am curious whether all types of expectations and norms (X) fall in the sociocultural norm category, or if some of them are infused with moral considerations to begin with. Put another way, I am curious if the authors can tease apart the specific moral and normative components within each functional expectation. Perhaps both types end up predicting moral judgment in unique ways (which would be very interesting), although my hunch is that the moral expectations are much stronger predictors of moral judgment than the sociocultural ones. Both friendship and teammate relationships may be hurt by Reciprocity violations, but perhaps friend relationships are hurt because the expectations are moral, while teammate relationships are hurt because the expectations are sociocultural. In the authors' work, which functional expectations are sociocultural, and which are moral? My guess would be that certain elements of Care, Reciprocity, and Hierarchy are moral to begin with, as they map onto moral dimensions in other lines of research (e.g., Moral Foundations Theory). But is Mating moral? Is "turning down someone for a date" a moral violation or is it a sociocultural violation, in the context of romantic partners? Similarly, Reciprocity expectations like "deciding not to return a nice favor" may be immoral in the context of close friendships, but amoral in the context of housemates or neighbors. Collapsing across social and moral norms poses a problem conceptually, but it also poses a problem analytically when it appears based on the

Sample 2 density plot data that there is little difference between siblings and teammates, housemates, bosses, or strangers. This distinction may help the authors clarify the null association they reported for the Reciprocity function. The authors suggested that this could be explained by the difference between exchange/transactional and communal relationship norms, which is reasonable, but perhaps other explanations are also possible.

We thank the reviewer for raising this very important point. In fact, we considered this possibility when we designed our studies, which led to the inclusion of an additional variable. Specifically, in addition to measuring participants' moral judgments of actions, we also measured their evaluations of how common/uncommon the actions were. We planned to control for this variable in our models in our pre-registered analysis plan. In the revised manuscript we now highlight how this aspect of our study design helps us to disentangle the effects of genuinely moral vs. social-conventional norms.

So, on page 14 of the revised manuscript, we now clarify the distinction between relational (moral) norms and social-conventional norms, citing Turiel's seminal work on this distinction; and we raise the issue of their relative contributions to explaining variance in moral judgments. In particular, we draw attention to the "action likelihood" variable that we measured, which should account for *any* deviation from what is expected, whether that expectation is moral in nature or sociocultural/conventional. Thus, if we control for the effect of action likelihood on moral judgments, the remaining variance captured by the "action likelihood" variable should plausibly reflect what is due to moral expectations that don't pertain to sociocultural/conventional expectations.

Now, as can be seen in our currently reported regression equation (the full table of results is Table S11f in the Supplementary Materials; partially reproduced below), both relational norms and action likelihood significantly predict moral judgments. Thus, consistent with the reviewer's intuitions, the sheer unlikelihood of an action (i.e., in violating a social norm, moral or otherwise) does indeed explain some of the variance in moral judgments in our study. However, as can be seen, the beta value for relational norms (16.26) is 80 times larger than that for the action likelihood ratings (-.20) when both are included in the same model:

S11f. Full regression table for main analysis, controlling for demographic information.

Predictor	β	Std. Error	95% CI [LL, UL]	p
(Intercept)	76.03	1.31	[73.45, 78.60]	< .001
Relational Norms	16.26	0.32	[15.64, 16.88]	< .001
Action Likelihood	-.20	.01	[-.21, -.18]	< .001
Target Specificity	0.37	.01	[.34, .40]	<.001

To further explore the relative contributions of relational norms versus social-conventions in predicting moral judgments, we ran additional models which we now report in the paper. As we now state in the manuscript (p. 14 of the revision):

The “action likelihood” variable serves an additional, theoretically important purpose. As we alluded to previously, it can help account for the variance in moral judgments that is due to potentially non-moral violations of social-conventional expectations (i.e., deviations from what is socially expected, whether or not the expectation tracks a perceived moral obligation) (30) as opposed to violations of relational norms specifically. By comparing the R^2 effect size estimates and AIC goodness-of-fit scores (i.e., of relational norm versus action likelihood models) we can judge the relative impact of each metric in explaining moral judgments across relationships. We find that, in a model with no information about relational norms, action likelihood alone does significantly predict moral wrongness judgments in the absence of other predictors ($p < .001$). However, this model explains much less variance, with a poorer goodness-of-fit score (marginal $R^2 = .08$, AIC = 136,496.9) than a model based only on relational norms (marginal $R^2 = .30$, AIC = 130,804). Moreover, the beta value for relational norms (16.26) is more than 80 times larger than that for the action likelihood ratings (-.20) when both are included in the same model (see Table S11f in *S/ Appendix*). This shows that relational norms explain moral judgments in this study far better than do merely conventional norms regarding what is socially expected.

R3.2. Clarifying how our findings relate to prior work on, e.g., utilitarianism vs. deontology

The authors are absolutely right to note that prior research about moral judgment toward “faceless strangers” does not apply to most of the

morally-relevant interactions people have with others in their social networks. However, that prior work has yielded many insights that the authors may benefit from incorporating into their analyses, or at least some suggestions future work. For instance, some individuals are more prone to perceiving something as moral because it maximizes welfare for the most people even if minor harms are caused (utilitarianism) or because it follows prescriptive rules (deontology) or because of downstream aftereffects (consequentialism). How do these inform moral judgments in social contexts? Do people rate close friends/family members high in Care or Reciprocity because of deontological norms or because of utilitarianism? Or do these have nothing to do with each other?

We are pleased that the reviewer brought up the possibility of individual differences in people's propensity to adhere to different meta-ethical principles (e.g., utilitarianism or consequentialism vs. deontology). Although empirically pursuing this question goes beyond the scope of the present paper, we are currently collecting data for a separate project in which we do in fact measure individual differences in these propensities to see how they affect both participants' prescribed cooperative functions for different relationships, and their moral judgments of wrongness for neglecting or thwarting those functions. In addition, the reviewer's comment has prompted us to engage in further thinking about how our present results relate to the large body of prior work on utilitarian versus deontological judgments. Accordingly, we have added material to the discussion section bringing out several points (pp. 19 of the revision):

Much of the prior literature in moral psychology has focused on judgments of strangers involved in moral dilemmas that pit distinct ethical principles against one another: for example, a utilitarian imperative to maximize welfare, versus a deontological rule that forbids using individuals as a mere means to an end (31). A key tenet of utilitarianism is that welfare should be maximized *impartially*, rather than prioritizing the well-being of family members (for example) over distant strangers (32). Descriptive research on moral dilemmas shows that many people are not in fact impartial in this sense (33), consistent with our observations here that people have different cooperative expectations for different relationships, leading in turn to different moral judgments depending on

relational context. One intriguing possibility is that individuals who more strongly endorse impartial beneficence will have more uniform prescriptions for cooperative functions across relationships, leading to more uniform moral judgments across relational contexts. This perspective also suggests possible antecedents of impartial beneficence. Because care is normative in close relationships (with family, friends, and romantic partners), caring for partners in these relationships does not typically elicit special approbation. Perhaps those who find a sense of purpose or belonging not in tending to close relationships, but in widely being admired (34), tend to “distribute” care across a broader set of relationships (thus showing relatively impartial beneficence).

References in this document

1. Rai TS, Fiske AP. Moral psychology is relationship regulation: moral motives for unity, hierarchy, equality, and proportionality. *Psychological Review*. 2011;118(1):57–75.
2. Curry OS, Mullins DA, Whitehouse H. Is it good to cooperate? Testing the theory of morality-as-cooperation in 60 societies. *Current Anthropology*. 2019;60(1):47–69.
3. Haidt J, Joseph C. The moral mind: How five sets of innate intuitions guide the development of many culture-specific virtues, and perhaps even modules. In: Carruthers P, Laurence S, Stich S, editors. *The Innate Mind*. New York: Oxford University Press; 2007. p. 367–91.
4. Bugental DB. Acquisition of the algorithms of social life: a domain-based approach. *Psychological Bulletin*. 2000;126(2):187–219.
5. McManus RM, Mason JE, Young L. To whom are we most obligated? Moral values and genetic relatedness structure perceived obligations and their impact on moral judgment. *PsyArXiv*. 2020;pre-print.
6. Gilead M, David YB, Ecker Y. Not our fault: judgments of apathy versus harm toward socially proximal versus distant others. *Soc Psychol Pers Sci*. 2018;9(5):568–75.
7. Clark MS, Earp BD, Crockett MJ. Who are “we” and why are we cooperating? Insights from social psychology. *Behav Brain Sci*. 2020;43(e66):21–3.

8. Simpson A, Laham SM, Fiske AP. Wrongness in different relationships: relational context effects on moral judgment. *J Soc Psychol.* 2016;156(6):594–609.
9. Clark MS, Boothby E, Clark-Polner E, Reis H. Understanding prosocial behavior requires understanding relational context. In: Schroeder DA, Graziano WG, editors. *The Oxford Handbook of Prosocial Behavior.* Oxford University Press; 2015.
10. Bloom P. Family, community, trolley problems, and the crisis in moral psychology. *The Yale Review.* 2011;99(2):26–43.
11. Schein C. The importance of context in moral judgments. *Perspect Psychol Sci.* 2020;15(2):207–15.
12. Isern-Mas C, Gomila A. Naturalizing Darwall’s second person standpoint. *Integr Psych Behav.* 2020;online ahead of print.
13. Tomasello M. The moral psychology of obligation. *Behav Brain Sci.* 2020;43(e56):1–58.
14. Lee J, Holyoak KJ. “But he’s my brother”: the impact of family obligation on moral judgments and decisions. *Mem Cogn.* 2020;48(1):158–70.
15. McManus RM, Kleiman-Weiner M, Young L. What we owe to family: the impact of special obligations on moral judgment. *Psychological Science.* 2020;in press.
16. Everett JAC, Faber NS, Savulescu J, Crockett MJ. The costs of being consequentialist: social inference from instrumental harm and impartial beneficence. *J Exp Soc Psychol.* 2018;79:200–16.
17. Koleva S, Selterman D, Iyer R, Ditto P, Graham J. The moral compass of insecurity: anxious and avoidant attachment predict moral judgment. *Social Psychological and Personality Science.* 2014;5(2):185–94.
18. Marshall J, Mermin-Bunnell K, Bloom P. Developing judgments about peers’ obligation to intervene. *Cognition.* 2020;201:104215.
19. McGraw AP, Tetlock PE. Taboo trade-offs, relational framing, and the acceptability of exchanges. *Journal of Consumer Psychology.* 2005;15(1):2–15.
20. Selterman D, Koleva S. Moral judgment of close relationship behaviors. *Journal of Social and Personal Relationships.* 2015;32(7):922–45.

21. Selterman D, Moors AC, Koleva S. Moral judgment toward relationship betrayals and those who commit them. *Pers Relationship*. 2018;25(1):65–86.
22. Simpson A, Laham SM. Individual differences in relational construal are associated with variability in moral judgment. *Personality and Individual Differences*. 2015;74:49–54.
23. Sunar D, Cesur S, Piyale ZE, Tepe B, Biten AF, Hill CT, et al. People respond with different moral emotions to violations in different relational models: a cross-cultural comparison. *Emotion*. 2020;online ahead of print.
24. Tepe B, Aydınli-Karakulak A. Beyond harmfulness and impurity: moral wrongness as a violation of relational motivations. *Journal of Personality and Social Psychology*. 2019;117(2):310–37.
25. Waytz A, Dungan J, Young L. The whistleblower’s dilemma and the fairness–loyalty tradeoff. *J Exp Soc Psychol*. 2013;49(6):1027–33.
26. Weidman AC, Sowden WJ, Berg MK, Kross E. Punish or protect? How close relationships shape responses to moral violations. *Pers Soc Psychol Bull*. 2020;46(5):693–708.
27. Rowe SJ, Vonasch AJ, Turp M-J. Unjustifiably irresponsible: the effects of social roles on attributions of intent. *Social Psychological and Personality Science*. 2020;online ahead of print.
28. Mammen M, Köymen B, Tomasello M. Young children’s moral judgments depend on the social relationship between agents. *Cog Dev*. in press;
29. Sterri AB, Moen OM. The ethics of emergencies. *Philos Stud*. 2020;online ahead of print.
30. Turiel E. Thought about actions in social domains: morality, social conventions, and social interactions. *Cognitive Development*. 23(1):136–54.
31. Conway P, Goldstein-Greenwood J, Polacek D, Greene JD. Sacrificial utilitarian judgments do reflect concern for the greater good: Clarification via process dissociation and the judgments of philosophers. *Cognition*. 2018;179:241–65.
32. Mill JS. *Utilitarianism*. London: Parker, Son and Bourn; 1863. 122 p.
33. Kahane G, Everett JAC, Earp BD, Caviola L, Faber NS, Crockett MJ, et al. Beyond sacrificial harm: a two-dimensional model of utilitarian psychology. *Psychological Review*. 2018;125(2):131–64.

34. Hirsch JL, Clark MS. Multiple paths to belonging that we should study together. *Perspect Psychol Sci.* 2019;14(2):238–55.

REVIEWERS' COMMENTS

Reviewer #1 (Remarks to the Author):

The authors have addressed my concerns and I have no further issues with the manuscript. Again, I congratulate the authors on a great paper that I think will be a valuable addition to the literature.

Reviewer #2 (Remarks to the Author):

I think the authors have done a great job of responding to my and other reviewers' suggestions. I don't have any other comments or concerns. It's a very strong paper! Brent Simpson

Reviewer #3 (Remarks to the Author):

I am pleased to see that the authors carefully and fully addressed my comments and suggestions in their revised manuscript. I have no further recommendations beyond what I had in the original review.

Reviewer #1 (Remarks to the Author):

The authors have addressed my concerns and I have no further issues with the manuscript. Again, I congratulate the authors on a great paper that I think will be a valuable addition to the literature.

Thank you for your excellent comments on the manuscript and for helping us improve it.

Reviewer #2 (Remarks to the Author):

I think the authors have done a great job of responding to my and other reviewers' suggestions. I don't have any other comments or concerns. It's a very strong paper! Brent Simpson

We are really grateful for your positive comments, and for your earlier feedback.

Reviewer #3 (Remarks to the Author):

I am pleased to see that the authors carefully and fully addressed my comments and suggestions in their revised manuscript. I have no further recommendations beyond what I had in the original review.

Thank you – we are glad to learn that you are satisfied with how we addressed your previous comments.